# Protein-polymer bioconjugates via a versatile oxygen tolerant photoinduced controlled radical polymerization approach

Alexis Theodorou [1], Evelina Liarou [2], David M. Haddleton [2], Iren Georgia Stavrakaki [1], Panagiotis Skordalidis[1], Richard Whitfield [3], Athina Anastasaki [3✉] & Kelly Velonia [1✉]

The immense application potential of amphiphilic protein-polymer conjugates remains largely unexplored, as established "grafting from" synthetic protocols involve time-consuming, harsh and disruptive deoxygenation methods, while "grafting to" approaches result in low yields. Here we report an oxygen tolerant, photoinduced CRP approach which readily affords quantitative yields of protein-polymer conjugates within 2 h, avoiding damage to the secondary structure of the protein and providing easily accessible means to produce biomacromolecular assemblies. Importantly, our methodology is compatible with multiple proteins (e.g. BSA, HSA, GOx, beta-galactosidase) and monomer classes including acrylates, methacrylates, styrenics and acrylamides. The polymerizations are conveniently conducted in plastic syringes and in the absence of any additives or external deoxygenation procedures using low-organic content media and ppm levels of copper. The robustness of the protocol is further exemplified by its implementation under UV, blue light or even sunlight irradiation as well as in buffer, nanopure, tap or even sea water.

[1] Department of Materials Science and Technology, University of Crete, Heraklion 70013, Greece. [2] Chemistry Department, University of Warwick, Coventry CV4 7AL, UK. [3] Department of Materials, ETH Zurich, Zurich 8093, Switzerland. ✉email: athina.anastasaki@mat.ethz.ch; velonia@materials.uoc.gr

Protein-polymer conjugates are hybrid biomacromolecules designed to display the wide diversity of functional and structural characteristics of both their synthetic and biological component[1–3]. In practice, certain native proteins exhibit reduced stability under non-physiological conditions, are susceptible to enzymatic degradation, can trigger undesirable immune responses, cannot penetrate most biological barriers and are below the kidney filtration threshold[3–6]. Among other approaches, protein-polymer conjugates have mostly evolved to address such limitations and improve the stability, solubility, and biodistribution, increase circulation half-life and decrease antigenicity of such proteins[1,7,8]. In addition, the unlimited chemical diversity of polymers offers the possibility to convey enhanced properties to natural biomacromolecules. Since the first generation of PEGylated proteins several decades ago[9–12], hydrophilic protein-polymer bioconjugates of remarkable complexity and diverse functionality have evolved[8,13–19].

To expand the vast application potential of protein-polymer bioconjugates, a hydrophobic polymer moiety has been incorporated to further introduce self-assembling properties to the resulting amphiphilic bioconjugates. In the pioneering research introducing Giant Amphiphiles, Nolte and collaborators covalently attached hydrophobic polymers to proteins via diverse "grafting to" approaches[20–22] demonstrating a variety of self-assembled nanostructures, which were dependent both on synthetic protocol and molecular structure[23–25]. However, the conventional "grafting to" approach entails independent synthesis of sizeable orthogonal end-functionalized polymers, which are subsequently covalently attached to a large protein. Due to the intrinsic limitations of reactions between macromolecules, this approach is mostly efficient in bioconjugation with water-soluble and low molecular weight polymers. The grafting of hydrophobic polymers to proteins further necessitates use of organic cosolvents, which often trigger protein denaturation[26] and complicate purification. Inspired by ground-breaking research on protein-initiated ATRP by the groups of Lewis[27,28], Matyjaszewski[29], and Maynard[8,18,19], our lab reported the synthesis of Giant Amphiphiles based on a "grafting from" approach[30,31]. By design, this synthetic approach enables in situ encapsulation of hydrophilic (bio)molecules within the vesicular interior of the proteinosomes, as well as incorporation of hydrophobic compounds within their membrane, offering a significant advantage over conventional "grafting to" strategies[14,30–35]. Nonetheless, the use of conventional CRP (Controlled Radical Polymerization, also referred to as RDRP, Reversible Deactivation Radical Polymerization) systems necessitates deoxygenation that is typically achieved via high-cost, time-demanding processes such as freeze-pump-thaw and/or inert gas sparging, which require sophisticated equipment and can potentially cause protein denaturation or loss of enzymatic activity[6,36–38]. Recently, seminal work by the groups of Boyer[39,40], Matyjaszewski[41–43] and others[44–47] led to elegant approaches in which no external deoxygenation is required for the synthesis of well-defined polymers. Such systems typically employ enzymes (e.g., glucose oxidase)[48–50], sacrificial substrates[43], and/or increased concentrations of photocatalysts or reducing agents[51–53] to in situ consume oxygen prior to polymerization. However, the applicability of such systems in the synthesis of protein-polymer conjugates is currently limited, since external reducing agents and/or enzymes may significantly increase the complexity of the "grafting from" approach, affect protein integrity and render product isolation tedious.

Herein we present the first oxygen tolerant, additive-free photoinduced copper-mediated RDRP to graft hydrophobic and hydrophilic monomers from protein macroinitiators, introducing a general method for the synthesis of a variety of protein-polymer bioconjugates in quantitative yields (Fig. 1). Inspired by recent oxygen tolerant copper-mediated polymerization strategies[46,47], we present a photoinduced polymerization process, which eliminates the need for deoxygenation and provides a universal approach for the synthesis of protein-polymer bioconjugates from protein macroinitiators. Our methodology avoids damage to the secondary structure of the protein, is compatible with various proteins (e.g., BSA, HSA, GOx, beta-galactosidase) and monomer classes including acrylates, methacrylates, styrene derivatives, and acrylamides.

## Results

**Oxygen tolerant, photoinduced grafting of styrene from BSA.** Bovine serum albumin (BSA, 66.5 kDa) was chosen as the model protein for the purposes of this study, due to its non-bridged cysteine residue (Cys34), which allows for the specific modification with one initiating group per protein[54,55]. The biomacroinitiator (BSA-Br, $I_o$) was synthesized via a bioorthogonal Michael addition of 2-bromo-2-methyl-propionic acid 2-(2,5-dioxo-2,5-dihydro-pyrrol-1-yl)-ethyl ester to Cys34 (Supplementary Figs. 1–6, Supplementary Information)[30,31]. Upon bioconjugation, the product BSA-Br ($I_o$) was enriched by extensive dialysis, characterized and stored at 4 °C (Supplementary Fig. 7).

Based on our previous studies on conventional ATRP grafting of styrene (monomer, $M_n$)[30,31], we reasoned that a stable monomer emulsion would be necessary for quantitative grafting in terms of macroinitiator consumption. We therefore selected a feed molar ratio of styrene to BSA-Br ($I_o$) 5000/1 as a starting-point value (Table 1, Entry 1). Importantly, all polymerization reactions were performed in widely used polypropylene syringes, where after addition of the reagents, headspace could be easily eliminated (Figs. 1, 2 and Supplementary Fig. 8). Upon judiciously optimizing the polymerization conditions, a ratio of $M_n/I_o/Cu^{II}/L = 5000/1/1.5/12$ was selected and all reagents were carefully loaded in a plastic syringe, which was subsequently exposed to UV irradiation (broad band ~365 nm, 36 Watt). The excess of the ligand Tris[2-(dimethylamino)ethyl]amine ($Me_6T$-REN) with respect to copper is essential for the in situ reduction of $CuBr_2$ to CuBr as previously reported[47,56]. Initially, a minor amount of organic solvent (DMSO, ~5%) was added to the buffer solution as this would allow efficient emulsification for cases where the monomer is solid rather than liquid (total organic content (OC) = 18%). Under these conditions (Table 1, Entry 1), the polymerization was allowed to proceed for a total of 3 h, during which the reaction mixture gradually became opaque. Aliquots were withdrawn by simply pushing the syringe plunger, thus avoiding disruption to the reaction system by minimizing oxygen exposure. The product formation (BSA-polystyrene, BSA-PS) and more specifically the initiator consumption was monitored by both native polyacrylamide gel electrophoresis (PAGE) and size exclusion chromatography (SEC). The final product was enriched by a simple dialysis step.

In native PAGE, biomacromolecule mobility depends on both the charge and the hydrodynamic size of the protein while, as observed in previous studies[20,30,43], biomacromolecules with an overall amphiphilic character do not migrate past the gel front. As shown in Fig. 2a and Supplementary Fig. 9, during the course of the reaction the concentration of BSA-Br ($I_o$) gradually decreased and a new, non-migrating band, attributed to the produced amphiphilic bioconjugate, appeared and gradually became predominant. This result was further supported by SEC (Supplementary Fig. 9) where after a short lag phase, BSA-Br was consumed and a product with progressively larger hydrodynamic volume was formed. In all cases, Refractive Index (RI) and UV traces were in good agreement. Nuclear magnetic resonance (NMR) also confirmed the grafting of polystyrene

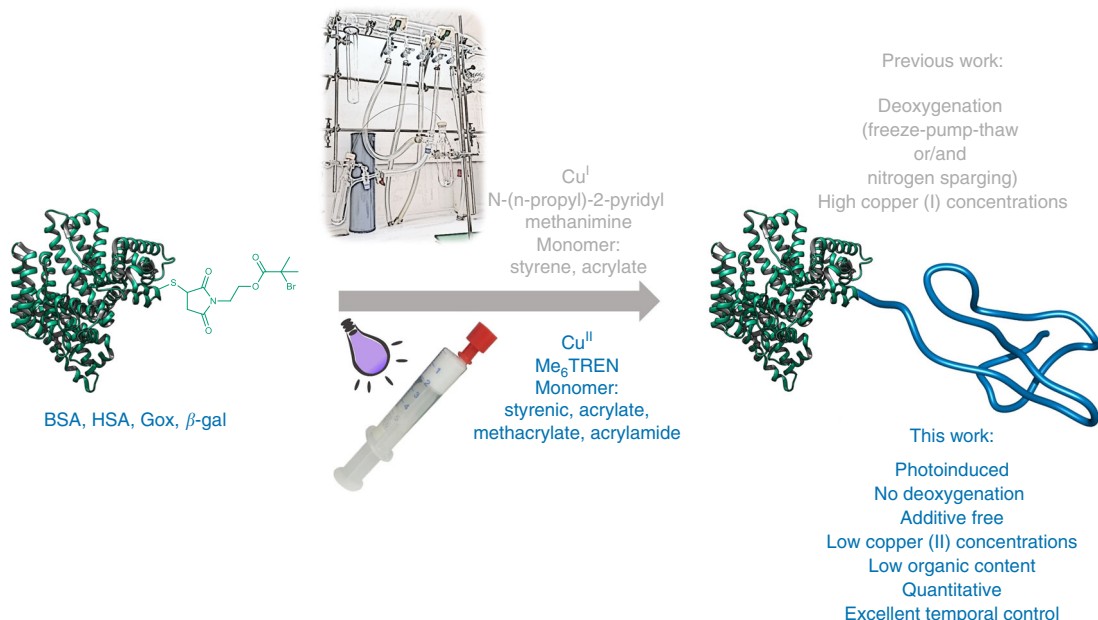

**Fig. 1 General scheme and setup for the synthesis of protein-polymer amphiphiles.** Top: Conventional ATRP approach and, Bottom: oxygen tolerant, photoinduced polymerization developed in this study.

**Table 1 Optimization of oxygen tolerant photoinduced ATRP grafting of styrene from BSA-Br.**

| Entry | $M_n/I_o/Cu^{II}/L$ | $Cu^{II}$ ppm | % OC | Rxn time min | $\lambda_{max}$ nm | BSA-Br ($I_o$) consumption |
|---|---|---|---|---|---|---|
| 1 | 5000/1/1.5/12 | 22 | 18[a] | 180 | 365 | Quantitative |
| 2 | 5000/1/1.5/12 | 22 | 13[b] | 180 | 365 | Quantitative |
| 3 | 5000/1/1.5/12 | 22 | 18[a] | n/a | 460 | Quantitative |
| 4 | 2000/1/1.5/12 | 22 | 11[a] | n/a | 365 | Quantitative |
| 5 | 2000/1/1.5/12 | 22 | 5[b] | 300 | 365 | Quantitative |
| 6 | 500/1/1.5/12 | 22 | 1[b] | 480 | 365 | Partial reaction |
| 7 | 50/1/1.5/12 | 22 | 0.1[b] | 480 | 365 | No or partial reaction |
| 8 | 5000/0/1.5/12 | 22 | 18[b] | n/a | 365 | No reaction |
| 9 | 5000/1/1.5/12 | 22 | 18[b] | 480 | dark | No reaction |
| 10 | 5000/1/0/0 | 22 | 18[b] | 480 | 365 | No reaction |

$M_n$ monomer (styrene), $I_o$ Initiator (BSA-Br), L ligand (Me$_6$TREN), OC organic content, Rxn time reaction time.
[a]Reactions in 20 mM phosphate buffer pH 7.4 with DMSO as cosolvent.
[b]Reactions in 20 mM phosphate buffer pH 7.4 without organic cosolvent.

(Supplementary Fig. 10A). Importantly, identical results and full consumption of the macroinitiator were observed in the absence of DMSO where the OC in the reaction medium (phosphate buffer) was limited to the amount of monomer and thus decreased to 13% (Table 1, Entry 2). The polymerization was performed under the same conditions using a single UV lamp (broad band ~365 nm, 9 Watt), leading to BSA-PS in high yield (Supplementary Table 1, Entry 18, Supplementary Fig. 11). UV light nevertheless might have detrimental effects on certain proteins via undesired photochemical reactions following diverse pathways and caused by the UV light excitation of the aromatic residues (Trp, Tyr, and Phe)[57]. For applications in which UV irradiation needs to be avoided, sunlight and blue light (460 nm) were also successful in the photoinduced grafting of amphiphilic bioconjugates in quantitative yields, highlighting thus the robustness of our approach to operate under milder conditions (Supplementary Table 1, Entries 12–15, Supplementary Fig. 11).

To explore the full potential of this system, different initial loadings of styrene ranging from 50 to 2000 equivalents were attempted. The reaction was found to proceed quantitatively at a feed molar ratio of styrene to BSA-Br of 2000/1 (Fig. 2a, b,

Table 1, Entries 4, 5, Supplementary Fig. 12). Importantly, with this monomer feed the total OC in the reaction mixture was as low as 5%. It should be mentioned that the total OC is a crucial, intrinsic parameter that needs to be independently evaluated for each protein in solution under defined conditions[58,59]. According to previous studies, BSA retains its native structure in the presence of low concentrations of DMSO (<10%)[59]. Partial or no consumption of the biomacroinitiator was observed when feeding molar ratios of 500/1 and 50/1 were used (Fig. 2a and Supplementary Fig. 13, Table 1, Entries 6, 7), underpinning our initial assumption that a stable emulsion is essential for quantitative grafting of hydrophobic monomers.

In order to validate these data and confirm the photoinduced nature of the polymerization, the reactions were systematically repeated in the absence of every single reagent. In the absence of either catalyst (copper/ligand) or BSA-Br macroinitiator no polymerization was detected by SEC or PAGE and the lack of polymerization was also evident in the dark (Supplementary Fig. 14, Supplementary Table 1, Entries 1–3). Since a relatively high ligand loading feed ($Cu^{II}/L = 1.5/12$, Table 1) was selected to ensure an excess of the tertiary amine would be available to

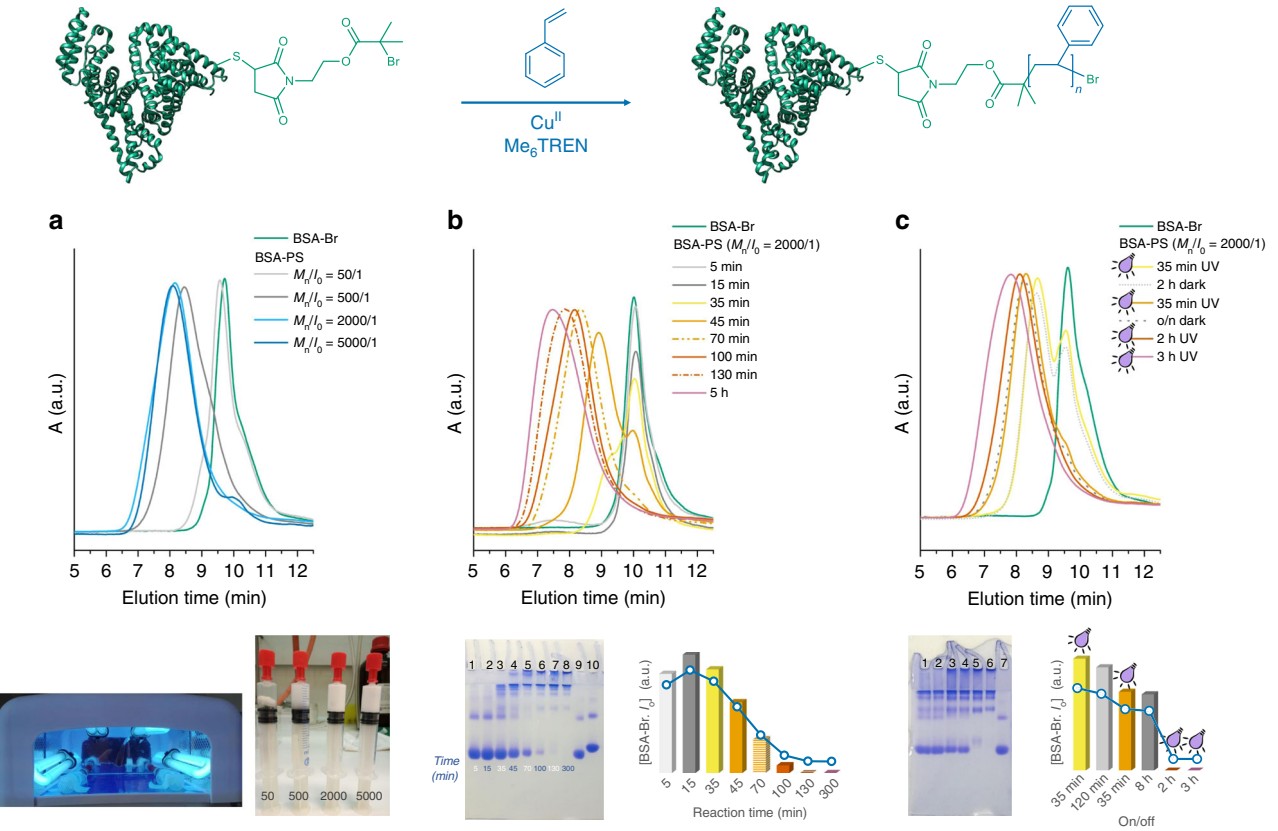

**Fig. 2 BSA-PS produced via oxygen tolerant photoinduced RDRP grafting from BSA-Br.** Schematic representation of the reaction. **a** Top: SEC chromatographs of bioconjugates formed during monomer feed ratio optimization, with [styrene]/[BSA-Br]/[Cu$^{II}$]/[Me$_6$TREN] ratios 50/1/1.5/12, 500/1/1.5/12, 2000/1/1.5/12, and 5000/1/1.5/12 (Table 1, Entries 2, 5–7); Bottom: Experimental setup and reaction syringes depicting the opaque solutions of the amphiphilic bioconjugates produced using increasing monomer concentration; **b** Time course of [styrene]/[BSA-Br]/[Cu$^{II}$]/[Me$_6$TREN] = 2000/1/1.5/12 reaction (Table 1, Entry 5), Top: SEC traces, Bottom left: Native PAGE lane 1: 5 min, lane 2: 15 min, lane 3: 35 min, lane 4: 45 min, lane 5: 70 min, lane 6: 100 min, lane 7: 130 min, lane 8: 300 min, lane 9: native BSA, lane 10: BSA-Br (I$_o$). Bottom right: Semiquantitative analysis plot of BSA-Br (I$_o$) consumption during the course of the reaction; **c** ON/OFF time course of [styrene]/[BSA-Br]/[Cu$^{II}$]/[Me$_6$TREN] = 2000/1/1.5/12 reaction (Table 1, Entry 5), Top: SEC traces, Bottom left: Native PAGE lane 1: 35 min ON, lane 2: 2 h OFF, lane 3: 35 min ON, lane 4: overnight (~8 h) OFF, lane 5: 2 h ON, lane 6: 3 h ON, lane 7: BSA-Br. The electrophoretic gel has been cropped for clarity. Bottom right: Semiquantitative analysis plot of BSA-Br (I$_o$) consumption during the course of the reaction.

mediate the reduction of the copper complex[47], addition of the complex caused a slight pH increase, though the final pH (7.66) did not affect protein stability. In agreement with our initial hypothesis, when a stoichiometric amount of ligand was used (1/1 Cu$^{II}$/L), the polymerization did not start verifying the necessity to utilize excess of the ligand (Supplementary Table 1, Entry 17, Supplementary Fig. 14).

**Temporal control**. The absence of polymerization in the dark encouraged us to assess the possibility of activating and deactivating polymerization by exposing the reaction mixture to alternating periods of UV light stimulation and darkness (Fig. 2c). As can be observed in Fig. 2c, the formation of BSA-PS rapidly occurred within the first 35 min of irradiation. However, under a prolonged period of ~8 h of darkness, no further biomacroinitiator consumption could be observed by PAGE while SEC revealed identical traces before and after the dark period. Importantly, when switching the light back ON, the polymerization could be reactivated and progress was monitored in the reaction. These results clearly indicate that the photoactivated grafting of monomers from proteins can be temporally controlled by simply regulating the UV light source toggling the reaction between ON and OFF states. This is in contrast to previously

reported solution polymerizations, which underwent long-lived linear growth during dark periods[56,60].

**Oxygen consumption**. To examine the oxygen consumption behavior of our system, we employed an in situ oxygen probe which enabled the online monitoring of the dissolved oxygen under the following conditions: [BSA-Br]:[Cu(II)]:[Me$_6$Tren] = [1]:[1.5]:[12] in water. Our experiments showed that in the absence of either the macroinitiator or the catalytic complex (copper and ligand combined), negligible, if any, oxygen consumption was detected within 2 h (Supplementary Fig. 10B). In contrast, in the presence of both the complex and the macroinitiator in aqueous solution, the amount of dissolved oxygen was significantly reduced, although not completely eliminated. The observed oxygen consumption was attributed to the in situ generated CuBr which subsequently abstracts the bromine yielding propagating radicals that may then react with the oxygen. The exact mechanism is currently under investigation. It is interesting to note that the polymerization starts before the full elimination of the oxygen, thus highlighting the oxygen tolerance of our approach. It should also be noted that the heterogenous nature of our system (i.e., emulsion) did not allow for oxygen probe measurements in the presence of monomer.

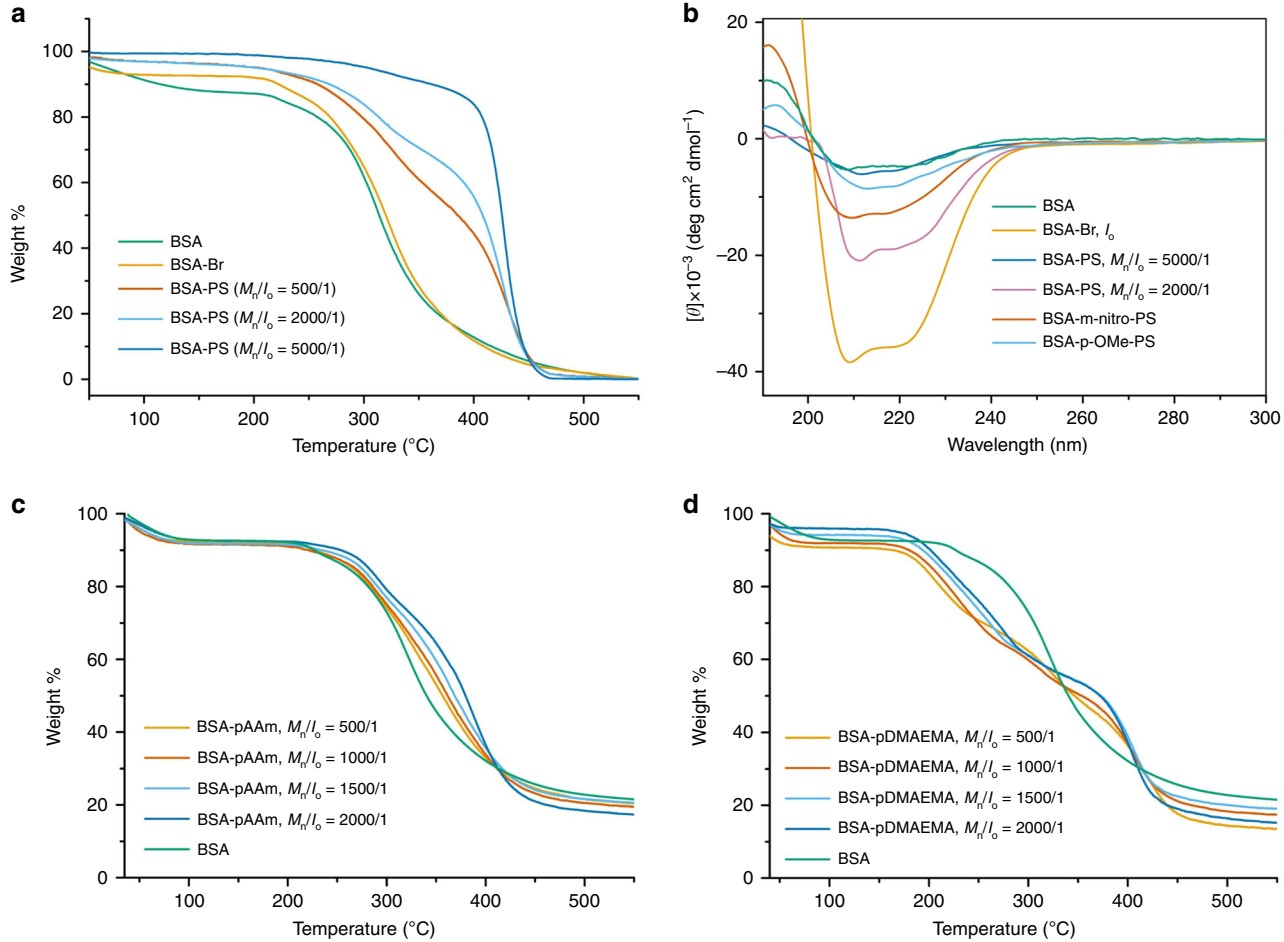

**Fig. 3 Structural characterization of BSA-polymer bioconjugates. a** Thermograms of BSA, BSA-Br, and BSA-PS conjugates ($N_2$ atmosphere). **b** CD spectra of BSA, BSA-Br ($I_o$), and BSA-polymer conjugates. (**c**) Thermograms of BSA, BSA-Br, and BSA-PAAm conjugates ($N_2$ atmosphere). **d** Thermograms of BSA, BSA-Br, and BSA-PDMAEMA conjugates ($N_2$ atmosphere).

**Structural characterization of BSA-PS giant amphiphiles**. Upon thermogravimetric analysis (TGA, Fig. 3a and Supplementary Figs. 13, 15), interesting trends were observed for the amphiphiles synthesized with different initial loadings of styrene. The control samples BSA and BSA-Br revealed similar behavior, i.e., an initial denaturation followed by decomposition over the same temperature range[61]. The thermograms of different amphiphilic BSA-PS bioconjugates (Table 1, Entries 2, 5, 6, Fig. 3 and Supplementary Fig. 13) revealed the same pattern at temperatures below 350 °C combined with a second significant steep weight loss at temperatures over 400 °C, which is attributed to the polystyrene moiety. Since TGA analysis can be utilized to semi-quantify the relative weights of the two components of block-copolymers we illustrate that the BSA-PS samples synthesized using lower initial loading of styrene (50 and 500 equivalents) led to the production of lower molecular weight PS while a further increase of PS content was observed when 2000 equivalents were used. Finally, the maximum amount of PS grafting was observed with 5000 equivalents while, identical TGA traces could be obtained from two different batch experiments thus highlighting the reproducibility of our approach (Supplementary Fig. 15).

To evaluate the effect of this grafting protocol on the protein, the secondary structure of BSA, BSA-Br, and BSA-PS were studied by circular dichroism (CD) in the far-UV spectral region (190–250 nm)[62,63]. Typically, BSA has a distinctive alpha-helical CD signature with positive ellipticity at ~193 nm and negative ellipticity at ~208 nm and 222 nm. As shown in Fig. 3b, BSA-Br

adopts the, distinctive for native BSA, α-helical structure as exhibited by the negative molar ellipticity at 222 nm and 208 nm, and the positive at ~193 nm. The helical conformation of the protein is evident for bioconjugates synthesized using a feed molar ratio of styrene to BSA-Br ($I_o$) of 2000/1 though a loss of the negative ellipticity is also observed. It is noteworthy that, when a feed molar ratio of 5000/1 was used, the loss of the negative ellipticity was more evident, indicating a decrease in the helical content. This might be attributed to the increased molecular weight obtained in the later grafting.

**Optimization studies**. For all the aforementioned experiments, a copper concentration of 22 ppm was utilized. However, we were able to further reduce the copper content to 6 ppm while achieving quantitative yields in terms of macroinitiator consumption which, to the best of our knowledge, is the lowest copper level ever reported for protein-polymer conjugates (Supplementary Fig. 12, Supplementary Table 1, Entry 8). A further decrease of the copper concentration below 6 ppm led to partial macroinitiator consumption, even after prolonged reaction times. To our satisfaction, the photoinduced grafting of styrene could also be quantitatively achieved by replacing the phosphate buffer reaction medium with either HPLC grade water (EC = 3 μS cm$^{-1}$ at 25 °C), tap water (EC = 442 μS cm$^{-1}$, pH 8.0 at 25 °C), or even sea water (EC = 47.2 mS cm$^{-1}$, pH 7.9 at 25 °C), which further highlights the robustness of this method (Supplementary Fig. 16, Supplementary Table 2, Entries 4, 5, 6).

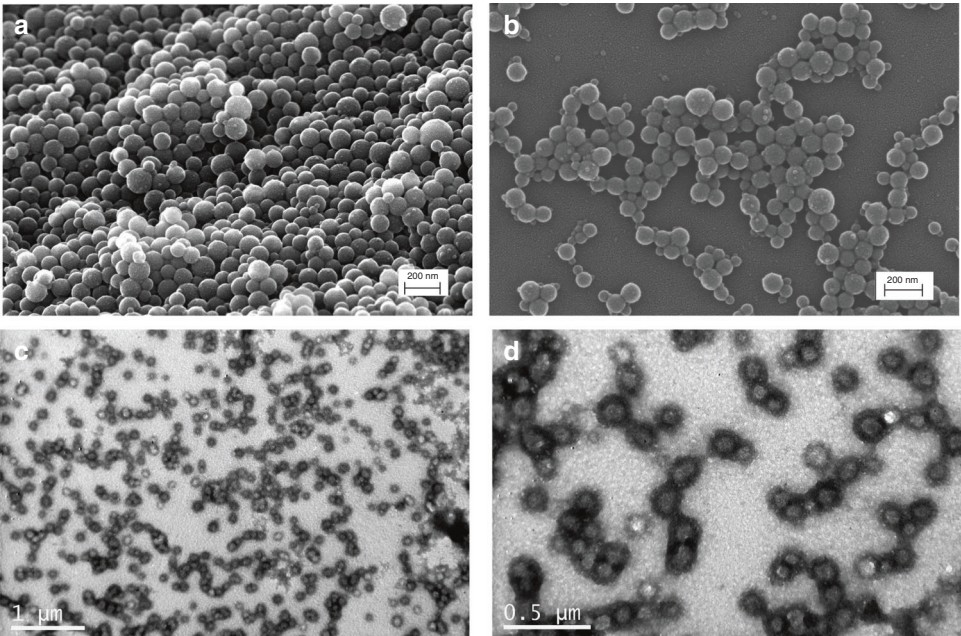

**Fig. 4 Morphological characterization of BSA-polymer bioconjugates.** SEM micrographs of BSA-PS uniform spherical assemblies synthesized using molar feed ratio **a** $[M_n]/[I_o]$=5000/1 and **b** 2000/1 respectively. **c**, **d** TEM micrographs of BSA-PDMAEMA spherical assemblies synthesized using molar feed ratio $[M_n]/[I_o]$=2000/1 (samples stained with uranyl acetate).

**Monomer range**. To address the efficiency of this oxygen tolerant approach in terms of monomer structure, the grafting of substituted styrenes, namely the *p*-methoxy (*p*-OMe) substituted and the *m*-nitro substituted styrene, from BSA-Br was studied at the optimum experimental conditions, i.e., using feed molar loading [substituted styrene]/[BSA-Br]/[Cu$^{II}$]/[Me$_6$TREN] = 2000/1/1.5/12. The formation of the amphiphilic protein-polymer hybrids was successful for both monomers as evidenced from both SEC and native PAGE (Supplementary Fig. 17). CD spectra (Fig. 2b) revealed no significant conformation changes for the *m*-nitro substituted BSA-PS amphiphiles, with the two α-helix negative bands being present, whilst exhibiting lower molar ellipticity. For the *p*-OMe substituted bioconjugate, the decrease in the negative molar ellipticity and the partial transition to a single negative band can be attributed to aggregation phenomena, related to β-sheet content and loss of the helical structure.

To investigate the versatility of this approach, we further studied the grafting of a range of hydrophilic monomers and various monomer classes, including an acrylamide (acrylamide, Am), an acrylate (oligo(ethylene glycol) methyl ether acrylate, OEOA$_{480}$) and a methacrylate (2-(dimethylamino)ethyl methacrylate, DMAEMA). In the case of Am, quantitative consumption of the BSA-Br macroinitiator was observed in PAGE when the optimized for styrene protocol was used at feed molar ratios $[Am]/[I_o] = 2000/1$ and 1500/1 (Supplementary Figs. 18 and 19). Traces of unreacted BSA-Br were observed at feed molar ratios $[Am]/[I_o] = 1000/1$ and 500/1. These findings were supported by TGA analysis of the products where the thermograms revealed the denaturation and decomposition of BSA at temperatures below 350 °C and a second significant steep weight loss at temperatures close to 400 °C, which is attributed to PAAm (Fig. 3c). Quantitative consumption of the BSA-Br macroinitiator was observed when the grafting of the acrylate was OEOA$_{480}$ at feed molar ratios $[OEOA_{480}]/[I_o] = 2000/1$ as evidenced by both PAGE electrophoresis and SEC (Supplementary Figs. 20 and 21). No high molecular weight products were observed at lower monomer feeds. The grafting of the water-soluble methacrylate

DMAEMA followed the trends of the hydrophobic styrene as all feeds above $[DMAEMA]/[I_o] = 500/1$ resulted in quantitative consumption of the macroinitiator as evidenced by PAGE (Supplementary Figs. 22 and 23). In fact, a viscous, opaque solution was gradually formed during polymerization, indicating that the resulting BSA-PDMAEMA conjugates were amphiphilic (Supplementary Fig. 24). PDMAEMA is a dually-responsive polymer with stimuli-responsive properties to pH and temperature that depend on molecular weight, structure, ionic strength, ion composition, and concentration[64]. These properties were expressed in the produced BSA-PDMAEMA both during polymerization and upon purification (Supplementary Fig. 24). The opaque solution of the enriched product (pH 7.4) became clear upon acidification with HCl at pH 5.0. In the TGA analysis of BSA-PDMAEMA, the characteristic weight loss attributed to the denaturation and decomposition of BSA were accompanied by weight loss, which could be assigned to the decomposition of the functional amino groups within the side chains of PDMAEMA at temperatures (close to 250 °C) and the degradation of its carbon skeletons (just above 450 °C, Fig. 3 and Supplementary Fig. 23)[65]. Interestingly, in contrast to Am and OEOA$_{480}$, DMAEMA grafting from BSA-Br was quantitative when using as low as 6 ppm of Cu$^{II}$.

**Self-assembly**. Scanning Electron Microscopy (SEM) and Transmission Electron Microscopy (TEM) studies revealed for all BSA-PS bioconjugates the formation of the expected[27–29] spherical superstructures with diameters between 80 and 130 nm which, using this synthetic approach, proved to be extremely well-defined and uniform (Fig. 4 and Supplementary Fig. 25). Similar structures were observed for the p-methoxy substituted styrene grafting products, while interestingly, the structures stemming from the *m*-nitro substituted styrene were more irregular and less defined (Supplementary Fig. 18). Spherical structures with diameters between 50 and 100 nm were visualized for BSA-PDMAEMA at pH 7.4 (Fig. 4 and Supplementary Fig. 24).

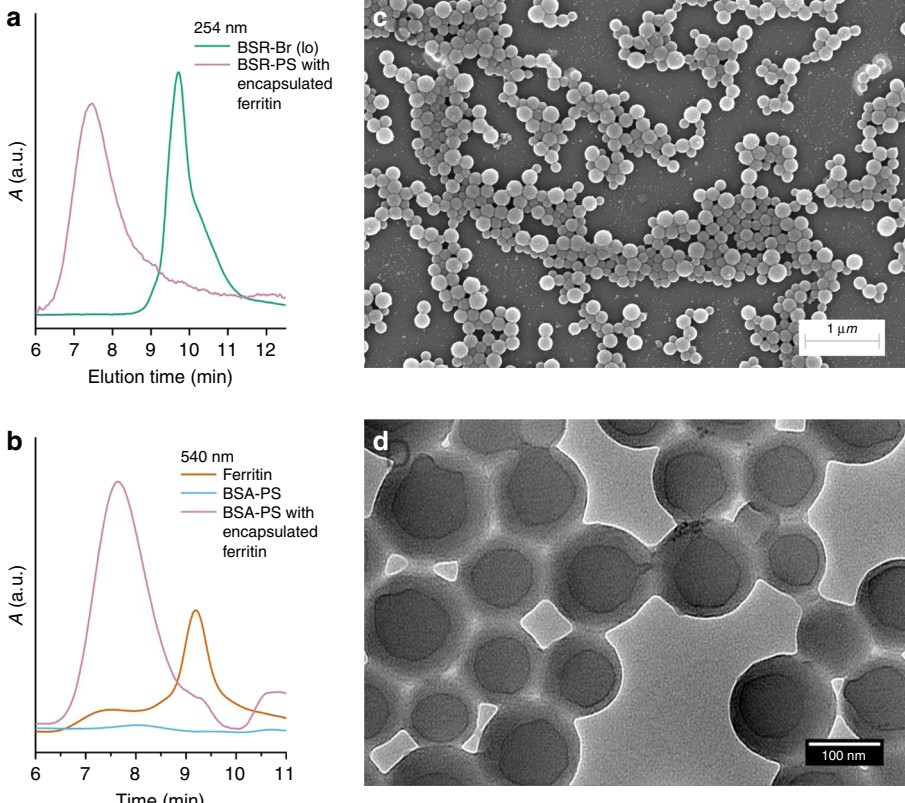

**Fig. 5 BSA-PS nanocarriers—ferritin encapsulation.** SEC traces at **a** 254 nm and **b** 540 nm. **c** SEM and **d** TEM micrographs of BSA-PS prepared in the presence of ferritin.

**BSA-PS nanocarriers**. Among the advantages of the grafting from approach for the synthesis of protein-polymer conjugates is the ability to in situ construct hierarchically assembled nanocontainers. Ferritins are ubiquitous in nature and, due to their unique structure, a subject of intense research for biotechological applications in fields varying from nanomedicine to biomineralization[66]. Iron-loaded ferritin is in fact widely used as a contrast agent in electron microscopy and MRI. To setup a basis for future applications, the grafting of styrene from BSA-Br was performed in the presence of ferritin in the reaction mixture. Upon completion of the reaction a supplementary precipitation step was added to remove the non-encapsulated ferritin. The product was analyzed by SEC, which at 254 nm verified the formation of a new product with larger hydrodynamic diameter (Fig. 5a). By setting 540 nm as detection wavelength, we directly confirmed ferritin encapsulation (Fig. 5a) as neither BSA, nor BSA-PS can be traced at this wavelength. The SEM and TEM micrographs (Fig. 5) revealed aggregation patterns similar to those observed for the polymerizations in the absence of ferritin. Both TEM and EDS analysis of the spherical nanostructures revealed the presence of ferritin (Supplementary Fig. 26). Since the encapsulation of ferritin during the formation of the biohybrid vesicles is expected to be statistical, the TEM micrographs revealed the presence of ferritin populations in both the interior and the membranes of the vesicles.

**BSA-PS esterase-like activity**. The effect of the photoinduced polymerization on the esterase-like activity of BSA was evaluated by monitoring the absorbance at 400 nm of the liberated p-nitrophenol produced from the BSA catalyzed hydrolysis of p-nitrophenyl acetate (pNPA, Supplementary Fig. 27). Both BSA-Br

and BSA-PS (Table 1, Entry 5) were found to retain part of the BSA esterase-like activity. The reduced activity can most probably be attributed to a change in the local environment around the active site upon grafting of the polystyrene or steric hinderance induced by self-organization. Similar results have been previously reported for denatured or partially unfolded state BSA[67].

**Applicability to other proteins**. To further expand the scope of our approach, the grafting from three more proteins was studied. Human Serum Albumin (HSA), is the most abundant translocator protein in blood circulation possessing critical physiological functions such as the maintenance of the colloidal osmotic blood pressure and the transportation of various endogenous and exogenous compounds. It has high structural homology to BSA (~75%) and bears one free cysteine that allowed to synthesize the HSA-Br macroinitiator via the bioorthogonal Michael addition of 2-bromo-2-methyl-propionic acid 2-(2,5-dioxo-2,5-dihydro-pyrrol-1-yl)-ethyl ester protocol used for BSA (Supplementary Information, Supplementary Figs. 28 and 29). Grafting of styrene from the HSA-Br macroinitiator proceeded via the oxygen tolerant photoinduced RDRP approach developed for BSA using a feed ratio of styrene/HSA-Br/Cu$^{II}$/L = 2000/1/1.5/12. The formation of HSA-PS was verified by the appearance of a new product with larger hydrodynamic volume in SEC and of a new, non-migrating band in PAGE attributed to the produced amphiphilic bioconjugate (Supplementary Fig. 29). The grafting of polystyrene under the selected conditions was judged to be quantitative as no initiator could be detected in SEC or PAGE. SEM imaging revealed the formation of uniform spherical superstructures with diameters between 80 and 100 nm for the HSA-PS bioconjugates (Fig. 6 and Supplementary Fig. 30). The

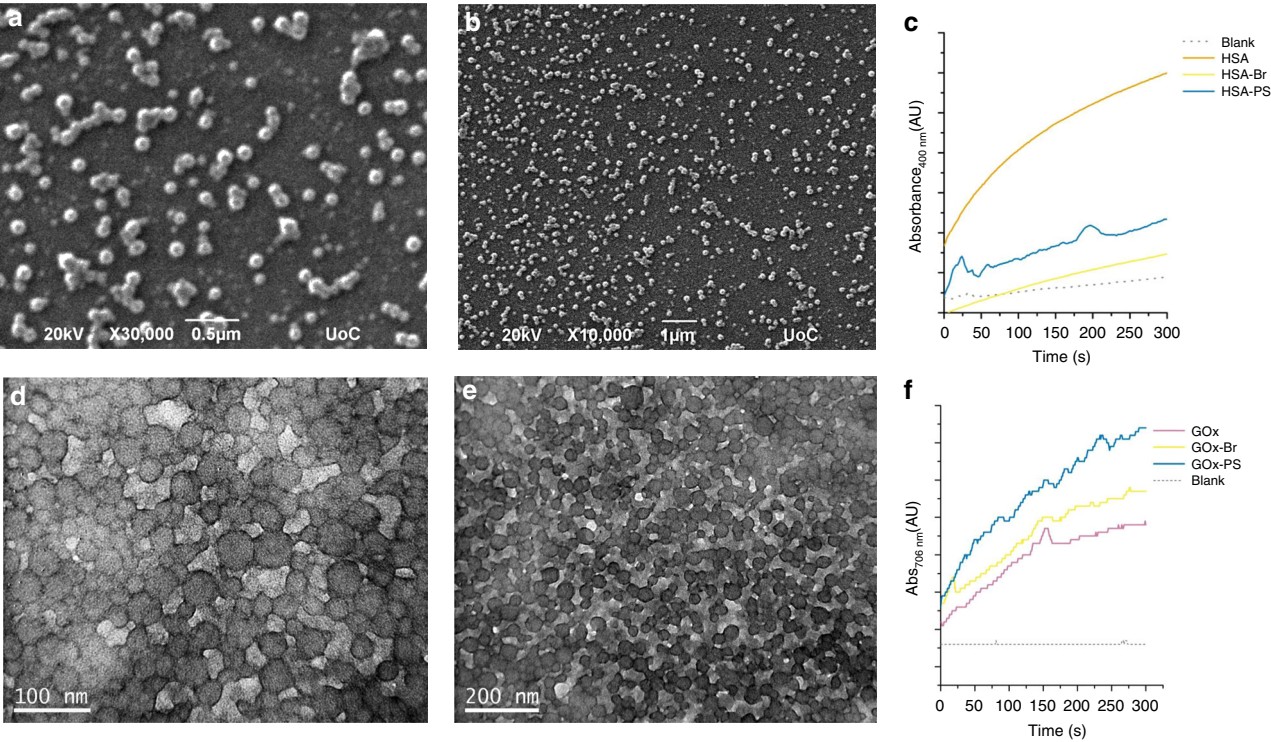

**Fig. 6 Morphological characterization and activity of protein-polymer bioconjugates. a, b** SEM micrographs of HSA-PS uniform spherical assemblies synthesized using molar feed ratio $[M_n]/[I_o] = 2000/1$; **c** Esterase-like initial activity of HSA and HSA-bioconjugates; **d, e** TEM micrographs of GOx-PS spherical assemblies synthesized using molar feed ratio $[M_n]/[I_o] = 2000/1$ (samples stained with uranyl acetate); **f** Initial activity of GOx and GOx-bioconjugates.

HSA-PS amphiphiles could catalyze the hydrolysis of pNPA following the same trends as BSA (Fig. 6c and Supplementary Fig. 29).

The beta-galactosidase from *Aspergillus oryzae* (β-gal, 120.6 kDa) is a monomeric enzyme that catalyzes the hydrolysis of lactose into glucose and galactose. Due to its hydrolytic activity on lactose and transferase activity on galacto-oligosaccharides production, β-galactosidase is widely used in food industry. The native protein bears four cysteine residues, none of which is readily available for bioconjugation. For this reason, we employed a modified literature protocol[68] to non-specifically link the RDRP initiator via NHS-ester coupling to exposed primary amines of the protein (Supplementary Fig. 31). Specifically, the biomacroinitiator β-gal-Br was synthesized via the slow addition of an equimolar quantity of N-hydroxysuccinimide-2-bromo-2-methylpropionate to the protein solution (Supplementary Fig. 32). Two different hydrophilic monomers, Am and DMAEMA were grafted from the protein macroinitiator using monomer feed ratio monomer/β-gal-Br/CuII/L = 2000/1/1.5/12 under the developed protocol conditions. Electrophoresis verified the formation of larger molecular weight bioconjugates through the new bands visualized for the products β-gal-PAAm and β-gal-PDMAEMA at and past the gel front. β-gal-PAAm was found to catalyze the hydrolysis of ortho-nitrophenyl-β-galactoside (ONPG) though with activity lower than the activity of the native enzyme (Supplementary Fig. 33).

Glucose oxidase from *Aspergillus niger* (GOx, 160 kDa) is a flavoprotein consisting of two identical subunits containing two tightly bound flavin adenine dinucleotides (FAD)[69]. GOx is a biologically and industrially important enzyme, widely used in food processing and gluconic acid production. To attach the RDRP initiator to GOX we followed the NHS-ester coupling protocol (Supplementary Figs. 34 and 35)[68]. Grafting of styrene from the GOx-Br macroinitiator proceeded under the developed protocol

conditions using a feed ratio of styrene/GOx-Br/CuII/L = 2000/1/1.5/12. In SEC a new peak with larger hydrodynamic volume appeared for the enriched product, while in PAGE a new, non-migrating band was observed while no biomacroinitiator could be detected. SEM and TEM imaging revealed the formation of spherical superstructures with diameters varying from 20 to 100 nm (Fig. 5 and Supplementary Fig. 36). To evaluate the effect of the grafting on GOx, a recently developed methodology entailing the enzyme induced formation of Prussian blue nanoparticles was employed as a simple colorimetric assay[70]. Interestingly, the GOx nanoparticles showed increased activity as compared to that of either the native GOx or the macroinitiator GOx-Br, suggesting that the grafting under UV did not affect GOx and, most possibly, the interaction with the polymer or the assembly itself enhanced protein activity (Fig. 6f and Supplementary Fig. 37).

In conclusion, using a commercially available, inexpensive UV light source (365 nm), the grafting of polystyrene from a BSA-macroinitiator in phosphate buffer reached quantitative conversion using extremely low levels of copper (as low as 6 ppm) without the need of inert gas sparging, freeze-pump-thaw or external additives. We developed an efficient polymerization grafting protocol, optimized in terms of reaction medium, catalyst concentration, monomer feed and light source and studied three different model styrenic monomers. Importantly we proved that during the in situ formation of such uniform, chimeric nanocontainers, the one-pot hierarchical incorporation of guest proteins is possible without steps that would interfere with the protein integrity or the overall architecture of the self-assembled superstructures. To expand the scope of this study, an acrylamide (Am), an acrylate (OEOA480) and a methacrylate (DMAEMA) were successfully grafted creating amphiphilic, hydrophilic and responsive protein-polymer bioconjugates. To access the universality of the approach, the grafting of monomers from three

more proteins with diverse in structural characteristics and functions (HSA, β-gal, and GOx) was demonstrated. All products were found to retain part of the catalytic activity of the native protein while, the grafting of styrene from GOx-resulted in increased catalytic activity. The results disclosed herein will significantly expand the availability of tailored polymer-protein bioconjugates and pave the way for future opportunities and directions.

## Methods

**Oxygen tolerant photoinduced synthesis of BSA-Br**. For the oxygen tolerant photoinduced RDRP grafting of styrene from BSA-Br using molar ratio ([styrene]/[BSA-Br]/[Cu$^{II}$]/[Me$_6$TREN] = 5000/1/1.5/12) in the absence or organic cosolvent, a solution consisting of styrene (0.25 mL, 2.185 mmol, 5000 equiv.) and nanopure water (0.3 mL) was initially sonicated for 30 sec to form an emulsion. Me$_6$TREN (14 µL, 52.44 × 10$^{-3}$ mmol, 120 equiv.) was added to 1 mL of a 1.5 mg/mL solution of CuBr$_2$ (6.55 mmol, 15 equiv.) in nanopure water to form a light blue colored solution due to the immediate copper-ligand complex formation. 100 µL of the CuBr$_2$/Me$_6$TREN solution (12 equiv. Me$_6$TREN and 1.5 equiv.CuBr$_2$, 0.34 mM) were added to the monomer emulsion and immediately transferred to a 5 mL syringe equipped with a stirring bar, containing a 0.32 mM solution of the BSA-macroinitiator (BSA-Br, I$_o$) in 20 mM phosphate buffer, pH 7.4 (1.25 mL, 0.437 × 10$^{-3}$ mmol). Headspace was eliminated to avoid the presence of undissolved oxygen and the reaction syringe was hermitically capped and placed under the UV or other light sources for specified amounts of time. Dialysis or removal of the monomer under reduced pressure preceded chromatography in all aliquots withdrawn from the reaction vessel for SEC and PAGE analysis.

For the oxygen tolerant photoinduced RDRP grafting of styrene from BSA-Br using molar ratio ([styrene]/[BSA-Br]/[Cu$^{II}$]/[Me$_6$TREN] = 5000/1/1.5/12) using DMSO as the organic cosolvent, a solution consisting of styrene (0.25 mL, 2.185 mmol, 5000 equiv.), DMSO (0.2 mL), and nanopure water (0.1 mL) was initially sonicated for 30 sec to form an emulsion. Me$_6$TREN (14 µL, 52.44 × 10$^{-3}$ mmol, 120 equiv.) was added to 1 mL of a 1.5 mg/mL solution of CuBr$_2$ (6.55 mmol, 15 equiv.) in nanopure water to form a light blue colored solution due to the immediate copper-ligand complex formation. 100 µL of the CuBr$_2$/Me$_6$TREN solution (12 equiv. Me$_6$TREN and 1.5 equiv.CuBr$_2$, 0.34 mM) were added to the monomer emulsion and immediately transferred to a 5 mL syringe equipped with a stirring bar, containing a 0.32 mM solution of the BSA-macroinitiator (BSA-Br, I$_o$) in 20 mM phosphate buffer, pH 7.4 (1.25 mL, 0.437 × 10$^{-3}$ mmol). Headspace was eliminated to avoid the presence of undissolved oxygen and the reaction syringe was hermitically capped and placed under the UV or other light sources for specified amounts of time. Dialysis or removal of the monomer under reduced pressure preceded chromatography in all aliquots withdrawn from the reaction vessel for SEC and PAGE analysis.

For the oxygen tolerant photoinduced RDRP grafting of styrene from BSA-Br using molar ratio ([styrene]/[BSA-Br]/[Cu$^{II}$]/[Me$_6$TREN] = 5000/1/1.5/12) in the absence or organic cosolvent, a solution consisting of styrene (0.10 mL, 0.874 mmol, 2000 equiv.) and nanopure water (0.45 mL) was sonicated for 30 sec to form an emulsion. Me$_6$TREN (14 µL, 52.44 × 10$^{-3}$ mmol, 120 equiv.) was added to 1 mL of a 1.5 mg/mL solution of CuBr$_2$ (6.55 mmol, 15 equiv.) in nanopure water to form a light blue colored solution due to the immediate copper-ligand complex formation. 100 µL of the CuBr$_2$/Me$_6$TREN solution (12 equiv. Me$_6$TREN and 1.5 CuBr$_2$ equiv., 0.34 mM) were added to the monomer emulsion and immediately transferred to a 5 mL syringe equipped with a stirring bar, containing a 0.32 mM solution of the BSA-macroinitiator (BSA-Br, I$_o$) in 20 mM phosphate buffer, pH 7.4 (1.25 mL, 0.437 × 10$^{-3}$ mmol). Headspace was eliminated to avoid the presence of undissolved oxygen and the reaction syringe was hermitically capped and placed under the UV or other light sources for specified amounts of time. Dialysis or removal of the monomer under reduced pressure preceded chromatography in all aliquots withdrawn from the reaction vessel for SEC and PAGE analysis.

All further experimental details and supplementary characterizations are provided in Supplementary Methods (Supplementary Figs. 1–39).

## Data availability

The authors declare that the data supporting the findings of this study are available within the article and its supplementary information files and from the corresponding authors upon request.

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

## Acknowledgements

The experimental work was supported by the Special Account for Research Funds of the University of Crete (KA 4726). The Special Account for Research Funds of the University of Crete (KA 3650) and Start up funds from ETH Zurich supported publication fees.

## Author contributions

K.V. and A.A. conceived the idea and designed the experiments. A.T. conducted the synthesis of all organic molecules and polymer-protein bioconjugates as well as the vast majority of the characterization including microscopy, NMR, MALDI, electrophoresis and GPC. E.L. and D.H. conducted the CD analysis and oxygen-probe measurements and interpreted the data in collaboration with K.V. E.L. conducted initial microscopy experiments and interpreted the data in collaboration with K.V. I.G.S conducted the TGA measurements. P.S. performed initial experiments for the project together with A.T. R.W. conducted MALDI-ToF-MS analysis and interpreted the data in collaboration with K.V. A.T., A.A., and K.V. analyzed all data with input from E.L. K.V. and A.A. co-wrote the paper. All authors commented on the manuscript. The overall supervision of the project was conducted by K.V.

## Competing interests

The authors declare no competing interests.
