## [Peer Review File · Nature Communications]

Reviewers' comments:

Reviewer #1 (Remarks to the Author):

The paper by Theodorou and colleagues presents exciting results in the synthesis of giant amphiphiles that contain proteins with hydrophobic tails. The work is very well described and if many issues are addressed could well be suitable for publication in Nature Communications. Indeed, if what is described in the paper is transferable generally to functional proteins, it could represent a real advance in the field. Most impressive is the elegant writing that tells the story effectively and presents the methods clearly.

This major issues with the paper will require some extra experimentation, but for a broadly focused journal the results described are too focused on a single non-functional structure. It is clear that the group and paper describes exquisite polymer science in generation a bioconjugate, but the biology/biotechnology side of the paper is not in the same league.

1. The paper describes the synthesis of BSA conjugates. The work is elegant but BSA is not a protein of general interest, and performing the work with one protein begs the question of whether this approach is viable with others. I recommend adding a table in which a variety (three?) of giant amphiphiles are synthesized and for which a simple activity test can be performed. This is the only general way to demonstrate retention of function.
2. The paper needs to describe more effectively why these data are not just an extension of the prior work and what makes the work completely novel. Are the authors sure, for example, that 21 ppm copper concentration is the lowest reported to date. The lowest Copper I concentration I have seen was about 0.1 mM (by the way, biologists far prefer using mM to ppm), and the lowest Cu II concentrations were 0.07 mM, I think. Both of these numbers would be less copper than 21 ppm.
3. Because BSA has no function, it is not clear what these particles are destined to be used for. Encapsulating ferritin is interesting, but do the authors propose these particles as a ferritin delivery vehicle? I suggest a much more careful description of how such particles could be used to accomplish a task that other approaches would fail at.
4. The synthesis of hydrophobic particles that contain a functional protein might open the door to effective non-aqueous synthesis. How would the authors expect these particles to behave in such systems?
5. Could these particles be useful in sensor applications?
6. The opening paragraph of the paper is a good demonstration of making extraordinary claims in order to inflate the importance of the work at hand. These claims are not supportable. The biomedical and biotechnological application of proteins is a \$150 billion business. Applications range from the best selling drugs to animal food and laundry detergent. The Nobel Prize was awarded this year for protein engineering. There is almost nothing in terms of protein function that cannot be achieved by directed evolution. Protein polymer conjugates are a tiny slice of the protein market (almost irrelevantly small outside of the therapeutic space). The authors should rewrite the entire introduction to account for the realities and in order to better explain where this new science fits in.
7. It is odd to include 19 references in a single citation. Perhaps the authors could reference some reviews and pick the most salient references that they need.
8. Although Maynard was one of the pioneers of the chemistry, she was not the first to perform protein-ATRP. That work was performed years earlier (and patented) by Andrew Lewis in the UK. The authors should properly reference Lewis' work. The next published work was from Matyjaszewski/Russell and then Maynard.
9. The oxygen removal strategies used in this paper are excellent. They do not need to be inflated by claiming that the addition of external reducing agents or enzyme for degassing is complex. Protein-ATRP is by now fairly routine chemistry.
10. Circular dichroism is not a high-resolution technique that allows one to determine if a protein has unfolded and lost function, which is why it is critical to use proteins that have measurable function.

11. What is the size of ferritin and how many iron ions were bound to it? Is there evidence as to where the ferritin is bound within the complex?

Overall, this is an elegant paper that presents potentially important data. The paper should be strengthened by stronger explanations of novelty (without resorting to inaccurate sweeping statements about the field), broader demonstrations that the chemistry retains structure (by using proteins whose function can be measured) and more careful/accurate descriptions of the current literature.

Reviewer #2 (Remarks to the Author):

The communication from Velonia and coworkers documents the efficient bio conjugation by an ATRP approach. The work is likely suitable for Nature Communications due to the nice oxygen tolerance. Having said that there are several things the authors need to address before I can be sure I can recommend the paper for nature communications.

A) The focus on giant amphiphiles is a nice application of the method, however, oxygen tolerance is a general problem in ATRP and other radical photochemistry. Indeed, the typical approach of freeze pump thaws and bubbling with inert gas can both lead to substantial enzyme denaturation. Could the authors run CD of their protein before and after these approaches? This does not need to be on the modified enzyme, but could be done on native BSA bubbled with N₂ or subjected to 3 freeze pump thaw cycles.

B) I think the authors need to demonstrate the oxygen tolerance using a range of monomers. Indeed, for many bioconjugations a hydrophilic polymer is far more likely to be used than a hydrophobic one like styrene. The authors should in their optimization table show. That their approach work for hydrophilic monomers such as DMAM, Am, OEOA etc.

C) The CD data show no qualitative shape change, but the authors should report this as molar ellipticity

D) There seems to be a lack of schemes used and description of why certain chemistries are chosen. For instance, more description and rationale for the OMe or nitro substituted groups are used. Is it a kinetic effect the authors are primarily interested in a self assembly or both?

E) How does the self assembly depend on the reaction extent? I am thinking of some of the PISA work pioneered by Armes and others.

F) Matyjaszewski and coworkers in 2012 showed that added bromide or chloride is important. The PBS can serve this role. Does the reaction still work with control in the absence of halide salts?

Overall, the work could be particularly impactful if it shows to work on a range of substrates such as acrylamide, or water soluble acrylates/methacrylates.

Reviewer #3 (Remarks to the Author):

This paper utilizes oxygen tolerant photo-ATRP to synthesize protein polymer conjugates with hydrophobic molecules. These amphiphilic biohybrids can form large biomacromolecular assemblies. The polymerization could be performed in polypropylene syringes using commonly available UV lights. I think the premise of the paper would be of interest to the audience of Nature Communications and is novel. However, the oxygen tolerance of the photo-ATRP process is the

lynchpin of that novelty. While the work is well done and thorough, there is insufficient discussion about the results, especially the oxygen tolerance. I strongly recommend that the authors expand the discussion and unpack the data, clearly showing what makes this so impactful. This may necessitate moving some information to the supplemental section. I recommend publishing after minor revisions.

Major Issues:

1. The discussion in the paper is insufficient, in particular with regards to the oxygen tolerance.
2. P1, L23 and P4, L141 - While the author states that other photoinduced RDRP methods do not have temporal control, insufficient evidence is given to this claim. While some methods may have better temporal control, this is not the first paper with temporal control in a photo-RDRP process.
3. The paper does not sufficiently indicate that the tertiary structure of the protein is still intact. This could be addressed with a method such as activity assays. If the thermograms of BSA (P17, Figure 2) are intended to be indicative of an intact tertiary structure, this should be discussed and more citations are necessary.
4. P3, L94 - 36 W UV light is strong. The potential issues with the use of UV should be included in the discussion.

Minor Issues:

1. P3, L78 - Please mention how much of the cysteine on BSA was modified with the initiator. Typically this is in the range of ~50%. When you are setting up your polymerization, this will affect your ratio of monomer to macroinitiator to copper, and that should be discussed.
2. P3, L96 - Total organic content is used frequently within the paper to describe the system. I presume that a sufficient amount is necessary for emulsification, but too much can have detrimental effects on the protein. I recommend elaborating on the desired range of total organic content.
3. P3, L92 - Please comment on the relatively high ligand loading that was used for this process. While this is addressed in citation 60, I think it would help the readers understand your process.
4. P3, L98 - I recommend putting the initial color of the reaction here as well as in the experimental
5. P4, L119 - Please include the molarities of the solutions as well as the molar equivalents. This is especially pertinent for the lower monomer feed ratios of styrene that were unsuccessful.
6. P4, L137 - Figure 1C and the dark experiment (figure S7) seem to indicate there is some consumption of the macroinitiator even in the dark.
7. P5, L196 - elaborate on the use of ferritin
8. P6, L219 - A relatively large amount of tertiary amine is used for this system. The buffering capacity of 20 mM phosphate buffer may be insufficient to account for this, and your pH may be much higher than 7.4. I recommend measuring your pH at the appropriate concentration and reporting it.

Minor issues in Figures:

1. Figure 1, A - The SEC traces for the conjugate with a DP of 2000 and a DP of 5000 look quite similar. Why don't you see a larger difference in molecular weight?
2. Figure 1, A - Please report how far the syringes are from the light source
3. Figure 1, B and C - please indicate how the data from the semi quantitative analysis plot was obtained.

Minor issues in Supplemental Information:

1. P3, L88 - I like that the UV sources are clearly separated into a section. Are the wattages measured or from the light bulb packages?
2. P8, L179 - For the synthesis of the BSA-macroinitiator, please put in the moles of each reactant used to improve the clarity.
3. P11, L 251 - Please label all of the lanes on the PAGE Gel
4. P14, Figure S13 - The label is cut off in C, and the

Reviewer #1 (Remarks to the Author):

The paper by Theodorou and colleagues presents exciting results in the synthesis of giant amphiphiles that contain proteins with hydrophobic tails. The work is very well described and if many issues are addressed could well be suitable for publication in Nature Communications. Indeed, if what is described in the paper is transferable generally to functional proteins, it could represent a real advance in the field. Most impressive is the elegant writing that tells the story effectively and presents the methods clearly. This major issues with the paper will require some extra experimentation, but for a broadly focused journal the results described are too focused on a single non-functional structure. It is clear that the group and paper describes exquisite polymer science in generation a bioconjugate, but the biology/biotechnology side of the paper is not in the same league.

We thank the reviewer for their kind support to our work. To make our contribution more broadly focused we have included 3 additional proteins and 3 additional monomer classes as shown below.

1. The paper describes the synthesis of BSA conjugates. The work is elegant but BSA is not a protein of general interest, and performing the work with one protein begs the question of whether this approach is viable with others. I recommend adding a table in which a variety (three?) of giant amphiphiles are synthesized and for which a simple activity test can be performed. This is the only general way to demonstrate retention of function.

This is a very good comment raised by the reviewer. Although BSA is a very commonly used protein in the field of polymer-protein bioconjugates, we took this opportunity to explore various other proteins that we could access. In summary, and in addition to BSA, we investigated the synthesis of polymer-protein bioconjugates from three additional proteins namely, human serum albumin (HSA), glucose oxidase (GOx) and beta-galactosidase. In the case of HSA, we bioorthogonally attached the same initiator as with BSA in order to obtain the required protein macroinitiator. Upon using our previously optimized conditions, we successfully polymerized styrene from HSA achieving quantitative macroinitiator consumption and yielding well-defined HSA-PS nanoparticles. Importantly, the activity of HSA (with the initiator attached) was retained upon polymerizing styrene, as confirmed by an additional activity test. In another example, GOx was also used as the protein moiety. Given the absence of accessible thiols in this protein (required to attach the maleimide-initiator), we synthesized and attached a NHS-bifunctional initiator (namely NHS). In a similar fashion to BSA and HSA, polymer-protein polystyrene giant amphiphiles could be obtained. Interestingly, the activity of GOx-PS was measured to be higher than the activity of the native protein. Beta-galactosidase was also modified with the NHS initiator although showing reduced activity upon polymerizing acrylamide. In addition to these three proteins, we were also interested in expanding the impact of our work by polymerizing different monomer classes. Apart from styrenic monomers included in our original submission, we have now demonstrated the compatibility of our methodology to three additional monomer classes including acrylates, methacrylates and acrylamides. Overall, our data show that our methodology is compatible to different proteins and monomer classes and as such can be of interest to a broad audience. To account for all these changes, the text in the manuscript has been modified accordingly.

2. The paper needs to describe more effectively why these data are not just an extension of the prior work and what makes the work completely novel. Are the authors sure, for example, that 21 ppm copper concentration is the lowest reported to date. The lowest Copper I concentration I have seen was about 0.1 mM (by the way, biologists far prefer using mM to ppm), and the lowest Cu II concentrations were 0.07 mM, I think. Both of these numbers would be less copper than 21 ppm.

We thank the reviewer for this comment. The concentrations have now been added using mM also in the Supporting Information.

The main conceptual advances of our work can be summarized below:

1) The oxygen tolerance of our methodology in combination with the facile nature of our conditions and the use of readily available reagents significantly expands the availability of tailored made polymer-protein bioconjugates to all researchers-both experts and not experts. By mixing everything together in vials or syringes and in the absence of any traditional deoxygenation procedures, well-defined bioconjugates can be obtained in a facile manner.

2) Our approach is compatible to different proteins and monomer classes including styrene, acrylates, methacrylates and acrylamides. To the best of our knowledge, this is the only methodology that allows for the polymerization of different monomer classes with both hydrophobic and hydrophilic monomers being successfully polymerized while demonstrating full macroinitiator consumption.

3) Other novel points of our approach include:

a) The use of very low copper concentrations

The reviewer encouraged us to search the literature very thoroughly with respect to the concentration of copper that we use. We found that we indeed use the lowest copper concentration in any traditional or photo-ATRP approach. We also found a paper published by Matyjaszewski's group using ICAR-ATRP where similarly low copper concentrations are used (*Macromolecules*, 2012, 45, 4461). This paper has been included in our references. In this publication however, no electrophoresis was shown and therefore the extent of macroinitiator consumption is unknown. This is an important piece of information, as we can employ much lower copper concentrations if full macroinitiator consumption is not required. Therefore, we are confident that our conditions use the lowest copper concentration possible while at the same time achieving full consumption of the macroinitiator.

b) Excellent temporal control

To the best of our knowledge, we demonstrate the first example of temporal control in the synthesis of bioconjugates. It should also be highlighted that with ATRP and even with simple polymerizations (rather than the synthesis of bioconjugates), excellent temporal control is not easily feasible due to the high copper concentrations and the high stability of CuBr in organic media. However, in our approach the use of low copper concentration and aqueous media lead to improved temporal control. This is because the small amount

of CuBr generated by the reduction of CuBr₂ in the presence of light, is immediately consumed in propagation events (due to the high K_p in water) and therefore has a limited lifetime which allows for significantly enhanced temporal control.

c) Quantitative macroinitiator consumption

Unlike many reported examples, our approach allows for all the macroinitiator to be consumed which improves overall efficiency and simplifies purification steps which remain non-trivial when the reaction mixture contains the native protein and amphiphilic bioconjugates.

d) The compatibility of our methodology with both UV and visible light

UV light, blue light or sunlight led to full macroinitiator consumption. In addition, the robustness of our method was further demonstrated by the use of either a buffer solution or nanopure water and was also efficiently operating in tap or sea water.

These conceptual advances have been further highlighted in the manuscript and abstract.

3. Because BSA has no function, it is not clear what these particles are destined to be used for. Encapsulating ferritin is interesting, but do the authors propose these particles as a ferritin delivery vehicle? I suggest a much more careful description of how such particles could be used to accomplish a task that other approaches would fail at.

Thank you for this comment. The encapsulation of ferritin is performed to demonstrate that these bioconjugates can be potentially used for applications involving encapsulation of cargos. Ferritin was selected as model because of its wide use in bionanotechnology (in fields spanning from nanomedicine to biomineralization) but also because of its high electron density that would allow its imaging in TEM. However, we would like to highlight that the novelty of our work is exclusively synthetic in nature. As mentioned in the previous comment, our methodology demonstrates a facile way to synthesize a wide range of polymer-protein bioconjugates. We very much appreciate this comment by the reviewer and we will seek to investigate such possibilities in a forthcoming publication.

4. The synthesis of hydrophobic particles that contain a functional protein might open the door to effective non-aqueous synthesis. How would the authors expect these particles to behave in such systems?

Though out of the scope of this work, we are also interested in the possibility of using giant amphiphiles in organic solvents. We have in fact already studied the BSA-PS giant amphiphiles in a variety of polar organic solvents. Interestingly, the self-assembled BSA-PS retains its spherical architecture and low diameter dispersity in both DMF and DMSO. In isopropanol and THF we could not image defined superstructures. We decided not to include this data in this manuscript as we would like to thoroughly investigate the possibility of performing such synthesis in organic solvents.

5. Could these particles be useful in sensor applications?

Polymersomes (and especially responsive polymersomes) are being intensively studied in sensor applications (Journal of Nanobiotechnology 16, 63, 2018) and as such, protein-polymer vesicles would also be useful. We are interested in investigating this application for a subsequent study, thank you for the suggestion.

6. The opening paragraph of the paper is a good demonstration of making extraordinary claims in order to inflate the importance of the work at hand. These claims are not supportable. The biomedical and biotechnological application of proteins is a \$150 billion business. Applications range from the best selling drugs to animal food and laundry detergent. The Nobel Prize was awarded this year for protein engineering. There is almost nothing in terms of protein function that cannot be achieved by directed evolution. Protein-polymer conjugates are a tiny slice of the protein market (almost irrelevantly small outside of the therapeutic space). The authors should rewrite the entire introduction to account for the realities and in order to better explain where this new science fits in.

We have made some changes to the introduction and toned down our claims.

7. It is odd to include 19 references in a single citation. Perhaps the authors could reference some reviews and pick the most salient references that they need.

We have narrowed down our reference list according to the reviewer's suggestion.

8. Although Maynard was one of the pioneers of the chemistry, she was not the first to perform protein-ATRP. That work was performed years earlier (and patented) by Andrew Lewis in the UK. The authors should properly reference Lewis' work. The next published work was from Matyjaszewski/Russell and then Maynard.

We agree with the reviewer, the suggested works have been cited in our manuscript and have been properly acknowledged in the text.

9. The oxygen removal strategies used in this paper are excellent. They do not need to be inflated by claiming that the addition of external reducing agents or enzymes for degassing is complex. Protein-ATRP is by now fairly routine chemistry.

We thank the reviewer for their comment. We would like to keep our points for not using external reducing agents or enzymes as we believe that this better highlights the simplicity of our method. We do not claim that the addition of reducing agents or enzymes is itself complex, we highlight that the process becomes complex due to several factors such as possible side reactions, incompatibility of many proteins with reducing agents and tedious purification of the final product (which would discourage many materials laboratories to use such systems).

10. Circular dichroism is not a high-resolution technique that allows one to determine if a protein has unfolded and lost function, which is why it is critical to use proteins that have measurable function.

As suggested by the reviewer we have now additionally performed activity tests for all 4 proteins and have modified the text accordingly. We thank the reviewer for this suggestion.

11. What is the size of ferritin and how many iron ions were bound to it? Is there evidence as to where the ferritin is bound within the complex?

Ferritin is a hollow, large globular protein which typically has internal diameters of around 8 nm and external diameters of about 12 nm. Within the spherical protein shell ferritin is able to accumulate and store up to 4500 iron atoms as superparamagnetic crystalline ferric oxyhydroxide. As mentioned above, one of the reasons ferritin was selected in this study was its high electron density that allows detection in TEM. The micrographs showed that ferritin was present both in the interior and on the membranes of the BSA-PS spherical structures (Figure S17, with a higher incidence in the membranes).

Overall, this is an elegant paper that presents potentially important data. The paper should be strengthened by stronger explanations of novelty (without resorting to inaccurate sweeping statements about the field), broader demonstrations that the chemistry retains structure (by using proteins whose function can be measured) and more careful/accurate descriptions of the current literature.

We would like to thank this reviewer for their support and for their important recommendations that have significantly enhanced the quality of our contribution.

Reviewer #2 (Remarks to the Author):

The communication from Velonia and coworkers documents the efficient bio conjugation by an ATRP approach. The work is likely suitable for Nature Communications due to the nice oxygen tolerance. Having said that there are several things the authors need to address before I can be sure I can recommend the paper for nature communications.

We thank the reviewer for their support.

A) The focus on giant amphiphiles is a nice application of the method, however, oxygen tolerance is a general problem in ATRP and other radical photochemistry. Indeed, the typical approach of freeze pump thaws and bubbling with inert gas can both lead to substantial enzyme denaturation. Could the authors run CD of their protein before and after these approaches? This does not need to be on the modified enzyme, but could be done on native BSA bubbled with N₂ or subjected to 3 freeze pump thaw cycles.

Following the reviewer's recommendation, we have added relevant literature that discusses how denaturation can occur. Those references have now been included in the manuscript. We also attempted to perform the suggested experiments ourselves and upon performing either freeze pump thaw cycles or nitrogen sparging, we observed the formation of significant amounts of foam which implies protein denaturation with the consequent losing of biological activity. Although we cannot study the foam, the literature conclusively attributes the foam formation in aggregates which have lost their activity. Additionally, BSA is a rather stable, robust protein, that easily refolds to its native state. Literature proposes studying structural changes of bovine serum albumin (BSA) under ultrasound treatment with Fourier transform infrared spectroscopy (FTIR) and fluorescence spectroscopy.

B) I think the authors need to demonstrate the oxygen tolerance using a range of monomers. Indeed, for many bioconjugations a hydrophilic polymer is far more likely to be used than a hydrophobic one like styrene. The authors should in their optimization table show. That their approach work for hydrophilic monomers such as DMAM, Am, OEOA etc.

This is a very important comment by the reviewer. We have now explored 3 additional monomers, namely OEOA (an acrylic monomer), DMAEMA (a methacrylic monomer) and Am (an acrylamide monomer). In the case of DMAEMA, well defined particles could also be obtained (although the monomer is water-soluble, the resulting polymer is not). OEOA and Am could also be successfully polymerized from BSA achieving quantitative macroinitiator consumption. We would like to thank the reviewer for this comment as it significantly expands the scope of our methodology which is compatible to a range of different monomer classes.

C) The CD data show no qualitative shape change, but the authors should report this as molar ellipticity

This has been amended, thank you for your suggestion.

D) There seems to be a lack of schemes used and description of why certain chemistries are chosen. For instance, more description and rationale for the OMe or nitro substituted

groups are used. Is it a kinetic effect the authors are primarily interested in a self assembly or both?

This is an interesting point raised by the referee. We were interested in both. Previous studies have shown that in ATRP of substituted styrenes, monomers with electron-withdrawing substituents resulted in better polymerization control and polymerized faster than those with electron donating substituent (Qiu, Matyjaszewski, *Macromolecules* 1997, 30, 19, 5643-564). However, electron donors at meta sites had surprisingly fast growth rates, which may be due to steric inhibition of termination. Nevertheless, we could not detect such an effect with our characterization techniques. Self-assembly was affected as shown in the relevant micrographs. The p-methoxy-substituted BSA-PS form smaller diameter spherical vesicular structures (and most probably also a population of micellar assemblies), than the m-nitro-substituted BSA-PS and non-substituted BSA-PS. Importantly in all cases the nanostructures are uniform. Perhaps more importantly, we were interested to assess the compatibility of our methodology to other monomers. In addition, to those styrenic monomers, we have now studied the polymerization of other monomer classes such as acrylates, methacrylates and acrylamides. Pleasingly, well-defined bioconjugates could be obtained in a facile manner.

E) How does the self assembly depend on the reaction extent? I am thinking of some of the PISA work pioneered by Armes and others.

This is a very interesting comment raised by the reviewer. In PISA systems, the macroinitiator is typically solvent-soluble but the monomer is initially solvent-soluble and becomes solvent-insoluble as the polymerization proceeds. However, in our case, as both hydrophilic and hydrophobic monomers can be polymerized, and to avoid confusion, we do not refer to our work as PISA.

F) Matyjaszewski and coworkers in 2012 showed that added bromide or chloride is important. The PBS can serve this role. Does the reaction still work with control in the absence of halide salts?

We thank the reviewer for this comment. The addition of bromide or chloride is not essential for our system. This can be demonstrated by the efficiency of our reaction in the absence of halide salts which has been shown when using either tap or HPLC grade water. This has been further highlight in the manuscript.

Overall, the work could be particularly impactful if it shows to work on a range of substrates such as acrylamide, or water soluble acrylates/methacrylates.

We thank the reviewer for their support and for the useful suggestion. Importantly, and following the reviewer's advice, we have now included 3 additional monomers which expand the scope of our methodology.

Reviewer #3 (Remarks to the Author):

This paper utilizes oxygen tolerant photo-ATRP to synthesize protein polymer conjugates with hydrophobic molecules. These amphiphilic biohybrids can form large biomacromolecular assemblies. The polymerization could be performed in polypropylene syringes using commonly available UV lights. I think the premise of the paper would be of interest to the audience of Nature Communications and is novel. However, the oxygen tolerance of the photo-ATRP process is the lynchpin of that novelty. While the work is well done and thorough, there is insufficient discussion about the results, especially the oxygen tolerance. I strongly recommend that the authors expand the discussion and unpack the data, clearly showing what makes this so impactful. This may necessitate moving some information to the supplemental section. I recommend publishing after minor revisions.

Major Issues:

1. The discussion in the paper is insufficient, in particular with regards to the oxygen tolerance.

We amended the discussion throughout the manuscript. Additionally, the oxygen consumption behaviour of our system was examined through *in situ* online monitoring of the dissolved O₂ concentration over time and these new findings were added and discussed in the manuscript.

2. P1, L23 and P4, L141 - While the author states that other photoinduced RDRP methods do not have temporal control, insufficient evidence is given to this claim. While some methods may have better temporal control, this is not the first paper with temporal control in a photo-RDRP process.

We apologize for the misunderstanding. Indeed, other ATRP polymerization approaches show temporal control. However, we could not find any example of temporal control that involves the synthesis of bioconjugates. In addition, all reported ATRP methodologies report imperfect temporal control. This is because they are mostly performed in organic media, where the lifetime of the in-situ generated CuBr (produced by the reduction of CuBr₂ in the presence of light) is high and therefore continues the polymerization even during the dark periods. However, our system can circumvent this by combining aqueous media with low ppm concentrations of copper. The low amount of copper significantly reduces the amount of produced CuBr and therefore improves temporal control. At the same time, in aqueous media, the produced CuBr is consumed significantly faster (higher *k_p* in water) and as such also improves temporal control. The combination of these two factors, lead to excellent temporal control and to the best of our knowledge, the first example of temporal control in bioconjugates made by ATRP

3. The paper does not sufficiently indicate that the tertiary structure of the protein is still intact. This could be addressed with a method such as activity assays. If the thermograms of BSA (P17, Figure 2) are intended to be indicative of an intact tertiary structure, this should be discussed and more citations are necessary.

Activity tests have been now included for BSA and for the additional 3 proteins examined, as requested by Reviewer 1. Thermogravimetric analysis was presented as an additional characterization methodology.

4. P3, L94 - 36 W UV light is strong. The potential issues with the use of UV should be included in the discussion.

We have amended the text accordingly-we would like to highlight that apart from UV irradiation, blue light and sunlight are also sufficient to afford the synthesis of well-defined bioconjugates.

Minor Issues:

1. P3, L78 - Please mention how much of the cysteine on BSA was modified with the initiator. Typically this is in the range of ~50%. When you are setting up your polymerization, this will affect your ratio of monomer to macroinitiator to copper, and that should be discussed.

The Michael addition of the maleimide-initiator (2-bromo-2-methyl-propionic acid 2-(2,5-dioxo-2,5-dihydro-pyrrol-1-yl)-ethyl ester) to Cys34 of BSA was performed using a large excess of the maleimide initiator over the protein (approximately 40-fold excess of the maleimide initiator over the protein) and the reaction was allowed to proceed for 48 hours. Under these conditions the reaction is quantitative. To prove this, we performed the same reaction using varying amounts of the initiator and proved that a larger than 20-fold excess ensures such a quantitative reaction. In a previous publication (*Angew. Chem., Int. Ed.* 47, 6263-6266, 2008), a titration using Ellman's reagent proved that, under these conditions, there are no available thiols post bioconjugation. We added this information and updated Figure S6 in the Supplementary information.

2. P3, L96 - Total organic content is used frequently within the paper to describe the system. I presume that a sufficient amount is necessary for emulsification, but too much can have detrimental effects on the protein. I recommend elaborating on the desired range of total organic content.

This is a fair comment raised by the reviewer. Organic solvents often alter the native structure of proteins by disrupting hydrophobic interactions between the nonpolar side chains of amino acids. On the other hand, numerous enzymes (such as lipases) are known to function in the presence of high organic solvent contents, and many in neat solvents or in supercritical fluids in the absence of water. The effect of organic solvent content on a protein depends both on the protein itself and the nature of the solvent. For BSA it has been previously shown that its native structure is retained in the presence of low concentrations of DMSO (<10%). The protein starts losing its structure with increasing amount of DMSO and attains a completely unfolded form in the presence of 40% DMSO. Since the intrinsic effect of the various monomers on the protein has not been reported, we optimized our reactions for minimum organic solvent content (in all cases less than 10%). Moreover, in blank experiments where the protein was incubated under the reaction conditions with the monomers (styrene, acrylamide, 5000 eq.), no effect was

observed in the protein migration on PAGE. To elaborate on the desired organic content in the manuscript the following phrase was added in P3, line 101:

... It should be mentioned that the total OC is a crucial, intrinsic parameter that needs to be independently evaluated for each protein in solution.* According to previous studies, BSA retains its native structure in the presence of low concentrations of DMSO (<10%).* ...

3. P3, L92 - Please comment on the relatively high ligand loading that was used for this process. While this is addressed in citation 60, I think it would help the readers understand your process.

We agree with the reviewer and have included a short explanation in the manuscript "The excess of ligand with respect to copper is essential to allow for the *in-situ* reduction of CuBr₂ to CuBr, as previously reported".

4. P3, L98 – I recommend putting the initial color of the reaction here as well as in the experimental

This has been amended, as suggested.

5. P4, L119 – Please include the molarities of the solutions as well as the molar equivalents. This is especially pertinent for the lower monomer feed ratios of styrene that were unsuccessful.

We have included exact concentrations, as suggested.

6. P4, L137 –Figure 1C and the dark experiment (figure S7) seem to indicate there is some consumption of the macroinitiator even in the dark.

In Figure 1C, we have performed a total of 3 on-off cycles. Indeed, in the first cycle a small degree of macroinitiator consumption could be observed in the dark. However, in the other 2 cycles no consumption was observed. Indeed, based on the chemistry employed, no macroinitiator consumption is expected in the dark. This is because the use of low copper concentration allows for a minimal CuBr production which is subsequently immediately consumed due to the high K_p in aqueous media. We thus attribute this small deviation in one of the measurements in experimental error during the measurement.

7. P5, L196 – elaborate on the use of ferritin

This has been elaborated, please see answer of point 3 by Reviewer #1.

8. P6, L219 – A relatively large amount of tertiary amine is used for this system. The buffering capacity of 20 mM phosphate buffer may be insufficient to account for this, and your pH may be much higher than 7.4. I recommend measuring your pH at the appropriate concentration and reporting it.

The pH was measured at slightly higher levels (7.66) and this has now been included both in the manuscript and the Supporting Information. Thank you for the comment.

Minor issues in Figures:

1. Figure 1, A – The SEC traces for the conjugate with a DP of 2000 and a DP of 5000 look quite similar. Why don't you see a larger difference in molecular weight?

SEC detects the assembled nanostructures and not the unimers, for this reason it cannot be used to access molecular weight difference. On the contrary, TGA gives this information through the % weight loss attributed to the polymer moiety, as compared to that attributed to the protein. A more systematic study to understand this further will be the subject of a forthcoming publication.

2. Figure 1, A - Please report how far the syringes are from the light source

This has been added in the Supporting Information, thank you for the comment.

3. Figure 1, B and C – please indicate how the data from the semi quantitative analysis plot was obtained.

The information has now been included in the methods section of the SI.

Minor issues in Supplemental Information:

1. P3, L88 - I like that the UV sources are clearly separated into a section. Are the wattages measured or from the light bulb packages?

All reported values are from the packages.

2. P8, L179 - For the synthesis of the BSA-macroinitiator, please put in the moles of each reactant used to improve the clarity.

This has been amended.

3. P11, L 251 – Please label all of the lanes on the PAGE Gel

This has been amended.

4. P14, Figure S13 – The label is cut off in C, and the legend is hard to read in B.

Thank you, it has been corrected.

REVIEWERS' COMMENTS:

Reviewer #1 (Remarks to the Author):

This revision is much improved. The authors are to be commended. I do think that the activity data are still a little scarce and not of the same quality as the polymer science. Perhaps the authors might create a Table or Figure for the main manuscript that summarizes functional data (or add it to the characterization figures).

For BSA and HSA, if I remember correctly, the esterase activity exists even in denatured protein. The authors should check this in a simple control...

After these simple additions this paper will be a tremendous addition to the literature.

Reviewer #2 (Remarks to the Author):

The authors have done an excellent job revising their manuscript and improving it to address reviewers concerns. I have no hesitation recommending the manuscript in its current form for publication after one minor revision.

The CD spectrum still has no axis label and so it is unclear if it is molar or specific elipicity. This could be addressed before this reviewer recommends publication

Reviewer #3 (Remarks to the Author):

I believe the authors have made a good-faith effort to address the previously raised issues of the reviewers. While not all of the suggested experiments were conducted, I think this is justifiable. This is, after all, a communication, and while it is possible to conduct an infinite number of additional experiments, I do not believe it is necessary.

The major scientific concerns and questions have been addressed. I believe this paper is ready for acceptance.

Reviewer #1 (Remarks to the Author):

This revision is much improved. The authors are to be commended. I do think that the activity data are still a little scarce and not of the same quality as the polymer science. Perhaps the authors might create a Table or Figure for the main manuscript that summarizes functional data (or add it to the characterization figures).

We thank the Reviewer for their comments.

The concentrations of the biohybrid amphiphile nanoparticle solutions used in the activity studies, were optimized to avoid scattering. The quality of the activity data is therefore greatly influenced by the low concentrations of the bioconjugates.

As similar trends were observed for BSA and HSA, and initial activity was difficult to be determined in beta-galactosidase, we added a figure summarizing the residual initial activity of two out of the four proteins reported in the manuscript (HSA and GOx), while all data are included in the Supplementary Information.

For BSA and HSA, if I remember correctly, the esterase activity exists even in denatured protein. The authors should check this in a simple control...

We thank the Reviewer for pointing this out, indeed we could trace residual activity in denatured BSA as previously reported in literature. A phrase was added in the manuscript, together with the relevant reference, now reading:

Page 7, line 25: Both BSA-Br and BSA-PS (Table 1, Entry 5) were found to retain part of the BSA esterase-like activity. The reduced activity can most probably be attributed to a change in the local environment around the active site upon grafting of the polystyrene steric hindrance induced by self-organization. Similar results have been previously reported for denatured or partially unfolded state BSA.⁶⁷

After these simple additions this paper will be a tremendous addition to the literature.

Reviewer #2 (Remarks to the Author):

The authors have done an excellent job revising their manuscript and improving it to address reviewers concerns. I have no hesitation recommending the manuscript in its current form for publication after one minor revision.

We would like to thank the Reviewer for this comment and evaluation.

The CD spectrum still has no axis label and so it is unclear if it is molar or specific ellipticity. This could be addressed before this reviewer recommends publication

We sincerely apologize for this -the label was mistakenly cropped out on the submitted version of the relevant figure. We added the uncropped graph.

Reviewer #3 (Remarks to the Author):

I believe the authors have made a good-faith effort to address the previously raised issues of the reviewers. While not all of the suggested experiments were conducted, I think this is justifiable. This is, after all, a communication, and while it is possible to conduct an infinite number of additional experiments, I do not believe it is necessary. The major scientific concerns and questions have been addressed. I believe this paper is ready for acceptance.

We would like to thank the Reviewer for their evaluation, we conducted as many experiments as were possible during this revision process. We sincerely believe that addressing the questions raised by all reviewers significantly improved both our understanding of this process and the quality of the manuscript.